# The New Testament and Workplace Mobbing: Structuring of Victims' Experiences

Jolita Vveinhardt [1,*] and Mykolas Deikus [2]

1   Department of Management, Faculty of Economics and Management, Vytautas Magnus University, 44248 Kaunas, Lithuania
2   Department of Theology, Faculty of Catholic Theology, Vytautas Magnus University, 44248 Kaunas, Lithuania
*   Correspondence: jolita.vveinhardt@vdu.lt

**Abstract:** Both practitioners and researchers confirm the utility of spiritual assistance for victims of violence, but the opportunities for religious spiritual assistance for persons who have experienced workplace mobbing have remained little explored in recent years. Although it is acknowledged that biblical narratives can help to structure personal experiences in coping with the traumatic consequences of violence, the main problem is the indefiniteness of the systematic application of specific texts in the process of assisting victims of workplace mobbing. In order to fill this gap, the analysis of the literature on workplace mobbing was performed and based on the identified essential features of the phenomenon, the types of response to violent behaviour in the Gospel of Luke were distinguished. Links between workplace mobbing and the gospel as well as guidelines for their practical application are discussed.

**Keywords:** workplace mobbing; violence; spiritual counselling; spiritual care; Gospel of Luke





## 1. Introduction

Research conducted in recent decades has demonstrated that the Christian approach to spiritual assistance offers an efficient response when coping with crises caused by traumatic situations (Jueckstock 2018). Spiritual help is efficient in alleviating psychological and physical sufferings (Sadat Hoseini et al. 2019), helping to recover after psychological traumas (Benson et al. 2016), finding answers to the questions related to personal identity (Utsch 2007), and existential questions (White et al. 2011). Ballaban states that the use of relating traumatic situations to biblical narratives serves to structure episodes of personal trauma, begins the process of integrating traumatic experiences, and initiates growth (Ballaban 2014). In addition, Rose suggests that the biblical conception of conflict, based on the plots described in the New Testament, can become a great example of the conflict management competence for local churches (Rose 2020). One of such radical conflicts, disintegrating one's personality and causing severe psychological and social consequences, is mobbing (Leymann 1996). This phenomenon is described as a long-lasting group attack on one person in the workplace, using verbal and non-verbal violence that causes strong emotional experiences, psychosomatic health disorders, destroys the personal, professional, and social life, and often ends in the victim's dismissal (Leymann 1996; Duffy and Sperry 2007; Nielsen et al. 2015; Pheko 2018). The usual assistance to the victims of mobbing is limited to psychological counselling in coping with stress, searching for ways out of conflict situations, or applying medication treatment (Stergiannis 2019). However, Woldemichael et al. (2013) carefully review traditional spiritual therapies used in psychotherapy. They note that both Christian pastoral care and psychotherapy use the most advanced human and natural sciences and represent important resources to maximise the understanding of a person, as well as increase the effectiveness of his/her care and treatment practice, but the biggest difference between these disciplines is related to their different anthropological views (Woldemichael et al. 2013). Therefore, the religious person who has experienced

violence at work finds significant support in inspiring biblical examples, which can be the basis for designing the response to a specific situation.

Sandnes (2016) notes that the Gospel of Luke is one of the most dramatic books of the Bible and has received increased attention among scholars. For example, he draws attention to the Gethsemane scene full of psychologism, which helps to understand the interaction between the Christian faith and the surrounding world. Other authors analyse examples showing the Christian response to injustice (Strom 2013), the relationship with persons who have experienced violence (Proctor 2019, p. 203), etc. However, so far, there is a lack of research analysing spiritual assistance to persons suffering from traumas caused by mobbing in the context of Christian pastoral care. That is, there is a lack of clarity about what kind of help is provided to the victims of workplace mobbing from various professional fields who turn to pastoral counsellors. For example, until now, several studies have been conducted focusing on mobbing in religious organisations (Nunez and Gonzalez 2009) and assistance to the clergy who have suffered from mobbing (Turner 2018). Vensel's (2012) research has also shown that some spiritual practices are applied to help to cope with the consequences of mobbing, but episodic excursuses to the topic of mobbing do not cover the opportunities for applying Christian spiritual help, seeking to help victims. This suggests that the efficient application of Christian spiritual assistance to the victims of mobbing requires broader conceptual-type research on biblical plots, which may serve in structuring the personal experiences of the victims of mobbing. Therefore, this study aims to typologize evangelical examples serving the purpose of structuring the personal experiences of the victims of mobbing.

First, this study draws a map of the mobbing conflict, distinguishing six typical categories of the process and its consequences. The methodology is presented and the content analysis is performed by distinguishing the plots of the Gospel of Luke, which illustrate the traumatic situations experienced by the victim of mobbing and the possible responses to them. Furthermore, we seek to answer the question: What non-confrontational ways of responding, based on the examples from the biblical text and empirical research, can be offered to the person suffering from workplace mobbing? Finally, conclusions and recommendations in providing spiritual assistance to the victims of mobbing are presented.

## 2. A Map of Workplace Mobbing

In an auto-ethnographic study, Pheko (2018) describes personally experienced mobbing as a conspiracy purposefully implemented by colleagues aimed at destroying his reputation and driving him out of work. Secret reports and letters were prepared "which contained fictitious incidents, incorrect statements, subjective evaluations, doctoring of minutes, professional character assassination and libellous insinuations, and presented them to the highest offices in the institution" (p. 4). The intensifying persecution was accompanied by a mass of unexplained symptoms including insomnia, nightmares, stomach aches, palpitations, anxiety, and excruciating physical pain, until finally he had to urgently go to hospital. The administration responded to the complaint about his colleagues' behaviour with baseless accusations and threats of dismissal, and slanderous rumours were targeted at the reputation of the victim as a scientist. The experienced distress also continued when trying to defend oneself in the courts. The story described by Pheko corresponds to a typical course of the process of workplace mobbing, which shows the desire to remove the victim from work, power imbalance, repeated intentional actions aimed at harming the victim, and the extension of the attack over time. Furthermore, these key features make up the map of workplace mobbing which will be briefly discussed.

Long-term persecution. As noted by Leymann, who is one of the pioneers of the research on workplace mobbing, in the industrialised Western world, the workplace is the only remaining "battlefield" where people can "kill" each other without fear of being convicted (Leymann 1996). He proposed to define mobbing as psychological terrorisation characterised by specific criteria of attacks: their frequency (at least once a week) and duration (attacks last for a period of not less than six months) (Leymann 1996). The phe-

nomenon is seen as various behaviours including persistent insulting and harassment, which cause extreme stress (Nauman et al. 2019). For example, Leymann singled out five directions of impact on the victim: limitation of opportunities for proper communication (communication opportunities limited by the management, verbal assaults, constant criticism, rejection, etc.); limitation of opportunities to maintain healthy social relations (colleagues stop communicating, communication is prohibited, the victim is moved to another room further away from co-workers, etc.); limitation of opportunities to retain personal reputation (background, appearance, state of health, gait, etc., are ridiculed); the impact on professional activities (tasks are not assigned or meaningless tasks are given, etc.); the effect on physical health (tasks dangerous to health are given, physical assaults, sexual harassment, etc.) (Leymann 1996; Zachariadou et al. 2018). During the process, the victim is pushed into the position where he/she feels helpless and unable to resist (Poirier et al. 2021).

Provocation and deceptive destruction. An essential feature of mobbing is social ostracism characterised by flawed dynamics of interpersonal relationships (Duffy and Sperry 2007; Sischka et al. 2021). Although Nielsen et al. (2015) note that it is difficult to provide a comprehensive list of mobbing actions, the mobbing process itself consists of two steps. First, the victim is systematically annoyed, and afterwards, subjective explanations of the victim's behaviour are provided. Since direct violence is not tolerated in modern organisations, deceptive actions are taken: aggressors can pretend to be victims themselves, making it difficult to identify them (Busby et al. 2022; Klein and Martin 2011). Thus, the victim may be provoked to react to the attackers aggressively, which is presented to the surrounding people as the victim's "bad qualities". According to Davenport et al. (1999), who divide the mobbing process more in detail into four phases, the victim is usually stigmatised in this way in the fourth and penultimate phase, before removing him/her from the organisation. Until then, the conflict is escalated in an ascending trajectory and is accompanied by the victim's mistakes. In the beginning, the conflict is not given prominence to, and later, the victim is provoked to attack, and in the course of the process, his/her psychological and physical state and reputation is only deteriorated in the eyes of observers (Davenport et al. 1999).

Group vs. person. The mobbing process always involves a group of persons (from one to four) who play different roles (Edmonson and Zelonka 2019) and who are linked by joint actions aimed at discreditation, humiliation, and social isolation (Matsson and Jordan 2022). According to Stalcup, this is due to the secret agreement (collusion) or because other members of the organisation look different (Stalcup 2013), as the discriminatory aspect of mobbing means that the victim is distinguished from the group due to personal, professional, or social characteristics inherent to him/her (Parkins et al. 2006). The victim is separated from the group, socially isolated, and due to the frequent and prolonged hostile behaviour, he/she experiences severe mental, psychosomatic, and social sufferings (Kowal and Gwiazda-Sawicka 2003).

Power imbalance. A group assault creates the imbalance of power that devastatingly affects the victim, making him/her feel powerless. However, a power imbalance is established even before mobbing, as their initiators seek to reinforce the experienced lesser personal power during the social persecution process (Duffy and Sperry 2007; Matsson and Jordan 2022). Additionally, Vandekerckhove and Commers (2003) suggest that mobbing often manifests itself as demonstrated power, failing to appropriately respond to the change and stress it causes. In this case, it can occur both horizontally (among employees at the same level) and vertically (the manager against a lower-ranking employee) (López et al. 2020). In any case, the violation of ethical norms (Matsson and Jordan 2022; Vandekerckhove and Commers 2003) and the belittling of the victim are aimed at social dominance and satisfaction as well as the elimination or widening of the power gap (Pheko 2018).

Seeking help. Three degrees of mobbing are distinguished: (i) the individual is able to resist, withdraws at an early stage, or is fully rehabilitated, (ii) the person can no longer resist, withdraws immediately, suffers from or experiences negative physical aftereffects,

(iii) the victim is no longer able to work and requires treatment (Davenport et al. 1999). In addition to the need for attention and love, Girardi et al. (2007) singled out the characteristic features of the victim such as depressed mood, difficulties in decision-making, suffering, and passive aggression (Girardi et al. 2007). According to Stadler (2006), the person who has experienced mobbing is in a clearly helpless position and usually cannot discontinue the mobbing without help from the outside (Stadler 2006). However, seeking help is aggravated by little support, isolation, and by the narrative of the constructed experience as a shameful personal failure (Duffy and Sperry 2007), as well as by social norms that construct the respective roles (Ross et al. 2021).

Removal of the victim. As already mentioned, the main aim of mobbing is to remove the victim from the organisation (Duffy and Sperry 2007; Khoo 2010; Pheko 2018; Leymann 1996). This process is described as constantly intensifying and strengthening, starting with individual attacks while the experienced pressure and threat grow until the victim is made redundant or withdraws himself/herself (Litzke and Schuh 2005). Conspirators acquire institutional power when they convince the leadership of the victim's blame, and the organisations themselves often contribute to conflict escalation and support the real perpetrators (Shallcross et al. 2010). Usually, the victim is removed from the organisation in the last phase of the mobbing process, and the dismissal is treated as the right decision in the organisation, continuing to blame the victim (Davenport et al. 1999; Hamre et al. 2021).

## 3. Materials and Methods

The study consisted of two stages. In the first stage, using the method of structural analysis of the scientific literature, categories and sub-categories are distinguished; in the second stage, the content analysis is performed. Considering the studied problem, the search for scientific literature included publications in the fields of theology and social sciences in international peer-reviewed scientific journals, databases using keywords such as "mobbing", "workplace bullying", "Christian spiritual assistance", and "New Testament". The analysis and systematisation of the selected scientific literature resulted in six categories and nineteen subcategories (Table 1).

**Table 1.** Categories and subcategories.

| Categories | Subcategories | Context |
| --- | --- | --- |
| Persecution | Lastingness<br>Verbal violence<br>Non-verbal violence<br>False accusations | The essential feature is the victim's persecution manifested by consistent verbal and non-verbal aggression and misconduct |
| Provocation | Situation-trap<br>Interpretation<br>Stigmatisation | The victim is provoked to act according to the intended plan, and upon receiving a confirmatory reaction, he/she is "marked" |
| Group/person | Conspiracy of the group<br>Search for the enemy<br>Social exclusion | Relationships between the group and the person are based on preconceptions, looking to detach the latter from the group's identity and separating him/her from it |
| Power imbalance | Threat of change<br>Direction of attack<br>Crowd dynamics | Restoration of imbalance of felt power as a threat of change, drawing vertical and horizontal attack directions, wherein crowd dynamics are exploited |
| Assistance | Internal dynamics<br>Institutional support<br>Circle of supporters | Sources of assistance, taking into account individual reactions from the institution and existing/potential supporters of the victim |
| Removal of the victim | Impact on the decision-maker<br>Institution's behaviour<br>Justification of the decision | The final stage and subsequent events, affecting the decision-making institute and efforts to justify the removal |

The ecumenical text of the Gospel of Luke, translated from ancient Greek into the researchers' native language and approved by the Lithuanian Bible Society and a foreign

consultant, was chosen for the content analysis. The text was also compared with the gospel written in the original language (accessed on 15 January 2022, https://www.greekbible.com/), as the literary translation does not always accurately reflect the intended meaning. The analysis was performed by two researchers who have practical experience in providing assistance to victims of mobbing: one is a representative of social sciences, and the other is a theologian with a minor in social sciences. While reading the text, the parts of it that recount the confrontations between Jesus and His opponents are singled out, and using the distinguished categories and subcategories of workplace mobbing (Table 1), the corresponding features and reactions to specific acts or attacks are distinguished. Reliability was enhanced through the repeated reading of the text and reflection. The results of both researchers' work were compared, discussed, and elements that did not correspond to the categories were rejected. Considering the categories that were formed, typical patterns of behaviour were distinguished while performing the content analysis. Questioning which elements of the analysed text corresponded to workplace mobbing by distinguishing categories and subcategories and the types of behaviour that emerge, context was also taken into account. The context in this case is important in answering the question of what circumstances the interpersonal interaction took place in and what the further development of the relationship was. The content itself is investigated from an etic perspective (Krysik 2018). According to Galdon, the biblical word τύπος corresponds to the meaning of the concepts' model or pattern and image (Galdon 1975), while typologisation is used as a method of interpreting the Bible (ibid.). This method is most commonly used when looking for the fulfilment of the prophetic signs of the Old Testament in the New Testament (Munk 1988), but biblical images and the human psychologism of Christ make it possible to transcend the frames of time (Johnson 1997), allowing for narratives of the Holy Scripture to sound modern (Decock 2008).

## 4. Features of Mobbing in the Gospel of Luke

Attacks on Jesus in the Gospel of Luke almost coincide with active activity. That is, from chapter 4 onward, when preaching begins and actions are taken, which provoke hostile religious reactions. Based on mobbing-specific acts of persecution, Table 2 distinguishes four subcategories, features of persecution, and Jesus' reactions.

**Table 2.** Features of persecution.

| Subcategory | Features | Reactions |
|---|---|---|
| Lastingness | Repeated attacks (Lk 5:21, 6, 2–7) and consistent observation (Lk 6:2). | Reasoned explanation instead of confrontation (Lk 6:3–5), preventive actions (Lk 6:8–10). |
| Verbal violence | Gossips (Lk 7:34), mockery (Lk 16:14; Lk 22:63–64; 23:35–37, 39), insulting (Lk 22:65), public ridicule (Lk 23:11), written mockery (Lk 23:38). | Highlighting of contradiction in gossips (Lk7:33–35), public unmasking (Lk 16:15), forgiveness (23:34). |
| Non-verbal violence | Frightening actions (Lk 4:29), causing of physical pain (Lk 23:33), coercion to perform humiliating "work" (Lk 23:26). | Dignified withdrawal (Lk 4:30), acceptance of shameful "work", giving it a symbolic meaning (Lk 23:27–31). |
| False accusations | Accusations of blasphemy (Lk 5:21), of sinful behaviour (Lk 5:30), of violation of law, traditions (Lk 6:2; Lk 15:2), of hostile identity (Lk 11:15), slandering (Lk 23:2, 5). | Public, reasoned explanation (Lk 5:22–24, 31–32; 6:3–5), arising doubt regarding the accusations (11:18–20), reference to the authority (Lk 13:15), forgiving (23:34). |

The New Testament is not an accurate diary of events, which would make it possible to restore a detailed chronology of the attacks, but the story of Jesus and his disciples shows a certain consistency and intensification of the attacks, which coincides with the trajectory

of the Journey to Jerusalem. For example, the conflict in Lk 4, 14–30 takes place with the community of Nazareth; while Jesus is continuing his activities, Pharisees and interpreters of the Holy Scripture, whose harassment distinguishes itself by more subtle forms (Lk 11:53–54), get involved, and ultimately, the secular government and religious leadership of Israelites become involved (Lk 13:31; Lk 19:47). Along with the involvement of secular and religious leadership (seen from chapter 19), the provocations and attacks increase. That is, the hostility is part of a snowball effect, and the way Jesus handles this and intends to act is discussed (Lk 19:47–48). It should be noted that accusations and slandering are intended to persuade the public and institutional authorities (Lk 23:2–5, 10), while non-verbal violence, which is first manifested in Nazareth, is intimidating and gives the impression of a threat to life. Jesus' reactions show that, on the one hand, he does not go into direct confrontation; on the other hand, going directly to the hostile crowd does not create the impression that he has been frightened (Lk 4, 30). In other words, it is not an approach that may encourage further persecution but rather a dignified and disarming reaction.

Provocation emerges as specific tactics aimed at forcing Jesus to act in a way that by, his actions, would confirm the pre-imposed blame. Meanwhile, the actions of Jesus himself prevent provocateurs from achieving their aim. Table 3 highlights the main features of his tactics.

**Table 3.** Features of provocation.

| Subcategory | Features | Reactions |
|---|---|---|
| Situation-trap | Questioning of "competence" (Lk 10:25), provocation (Lk 20:2). | Non-confrontational transfer of "responsibility" to the opponent (Lk 10:26), placing the opponents in an unsolvable situation (Lk 20 3:8). |
| Interpretation | Transforming the positive into the negative (Lk 11:15). | Comparison with opponents' supporters (Lk 11:19) and reasoning (Lk 11:21–22). |
| Stigmatisation | Marking (Lt 22:47), public equating with criminals (Lk 22:47, 52; 54), unofficial naming as a criminal (Lk 22:71), institutional and public noting of blame (Lk 23:26, 32). | Unmasking, naming of the action (Lk 22:48, 53), calm acceptance of reality (Lk 22:67–70), depersonalisation of the imposed image (Lk 23: 28–31). |

The answer to the Law Teacher's question (Lk 10:25–28) is an excellent model of a non-confrontational reaction to a provocative situation. The focus is first shifted from oneself to the authority (the Law), forcing the opponent himself to give the answer by giving the counter-directing question. The provocateur's energy is directed to himself, while praising and encouraging the provocateur himself to adhere to the declared principles make conflict escalation impossible. In general, it should be noted that when provocative situations forcing oneself to make excuses are created, Jesus never acts passively, and the reaction resemble micro-duels that neutralise the conflict escalated by the opponents. In addition, the imposed shame is peculiarly depersonalised, its meaning is extended to the symbol of the nation's suffering and its cause.

Positioning of the victim against the values and traditions uniting society also performs the function of unifying the hostile group and marks the victim as the enemy (Table 4).

**Table 4.** Features of building a hostile group–person relationship.

| Subcategory | Features | Reactions |
|---|---|---|
| Group conspiracy | Hostile discussion (Lk 6:11; Lk 19:47), coordinated, organised actions, espionage (Lk 20:20–21; Lk 22:2), betrayal (Lk 22:3–6). | Acceptance of reality, identification with the righteous who were persecuted and hopeful sense-making (Lk 6:20–23; 26), love vs. confrontation (Lk 6:35), forgiveness (Lk 6:37), critical self-evaluation (Lk 39:42), consistency (Lk 6:49), resistance to provocation (Lk 20:23–26), unmasking (Lk 22:21–22). |
| Searching for the enemy | Setting against the disciples of John and the Pharisees (Lk 5:33). Search for confirmation of a prior accusation (Lk 6:7), waiting for a mistake (Lk 11:53–54; Lk 14:1). | Reasoned explanation (Lk 5:34–39), insight of subterfuges (Lk 6:3–5), "forestalling the events" (Lk 14:3–5). |
| Separation | Non-acceptance (Lk 9:53), physical separation (Lk 22:54). | Identification with the righteous (Lk 6:23), refusal to take revenge/punish (Lk 9:55). |

It can be seen that the opposing group is consistently looking for mistakes and situations that would confirm the accusations; therefore, this insight hinders the escalation of the conflict. Jesus does not deny that his behaviour differs from the usual behaviour of certain groups and sometimes even emphasises this difference but does not go into direct confrontation and uses traditional wisdom-based reasoning in his comparisons. This way, the non-correspondence between the behaviour of the hostile groups themselves and their declared norm becomes evident. Such way of reasoning not only provides suggestibility but presents Jesus as a consistent continuator of the tradition and aggravates the opponents' attack. Blessings (Lk 6:20–23) are a key example of the response to the persecution of the hostile group, highlighting a humble acceptance of reality, giving a sense to it, and identifying with the righteous who were persecuted; that is, with the other group, which destroys the feeling of loneliness and of being separated and creates meaning.

The teaching that Christ and his followers triggered changes that threatened the long-established status quo of institutes governing the society and of their authority (Table 5).

**Table 5.** Features of power imbalance.

| Subcategory | Features | Reactions |
|---|---|---|
| Threat of change | Fear of the unknown (Lk 8:37), resistance to change (Lk 19:45). | Retreat by acting indirectly through works whose authorship is handed over to God (Lk 8:37), reasoning by authority/Law (Lk 19:46). |
| Direction of attack | Vertical: the group with the power of authority against the person (Lk 11:53–54), the institution against the person (Lk 13:31); horizontal: the crowd against the person (Lk 22:1; 66), increasing of power by persuading the decision-making institute (Lk 23:18–24). | Humble acceptance of reality without refusing one's identity (Lk 13:32–33). |
| Crowd dynamics | The majority (Lk 11:15–16), the "rebellious crowd" governed by emotions (Lk 23:18–23), passive approval of violence (Lk 23:35). | Doubt towards the accusation (11:18–20). |

On the one hand, actions that fall out of their context and their outcome cause fear and rejection, on the other hand, the power of the governmental institutions' authority is

embodied in the order and the support of the majority, which is hindered by Jesus' activities. It should not be forgotten that the power of both the senior Israelites and the religious leaders experiences a crisis due to domination of Rome. Power is symbolically restored by finding a common enemy, belittling, and gaining support from a higher authority (that has the decision-making power). Without the latter, the vertical trajectory of the attack would not be as efficient. The non-violent response aggravates the escalation of the conflict, and the humble acceptance of reality on the human plane protects from self-destruction, having encountered the insurmountable situation. In this context, a consistent line can be drawn between temptation in the desert (Lk 4:5–7), the twice-repeated warning to the disciples (Lk 22:40, 46), and sword symbolism in the event of arrest (Lk 22:50–51), which highlights an alternative relationship with reality.

Organised assault as a manipulation of the illusion of power pursues several goals: to create an impression of threat and to eliminate potential sources of support, both institutional and non-institutional, which deepens the victim's feeling of helplessness (Table 6).

**Table 6.** Behavioural characteristics and internal dynamics of the environment that can provide assistance.

| Subcategory | Features | Reactions |
| --- | --- | --- |
| Internal dynamics | The temptation to impose one's will on events (Lk 22:40,46), hope to avoid (Lk 22:42), agony, suffering [ἀγωνία] (Lk 22:44). | Prayer, devotion to the will of the Father (Lk 22:42–44), refusal of violence (Lk 22:51); forgiveness (23:34). |
| Institutional support | Absence of support, opting out (Lk 23:7). | A straight answer to allegations (Lk 23: 3). |
| Circle of loved ones | Withdrawal of a part (Lk 22:57–60), contempt for those persecuted together (Lk 23:39) and intercession (Lk 23:40–41), posthumous acknowledgement (Lk 23:47), being together (Lk 23:49; 52–56). | Naming of actions (Lk 22:61), fair reward for support (Lk 23:43). |

Experiencing helplessness and threat to life provokes aggressive reactions. Such reactions imply the temptation to accept the created illusory reality, may escalate violent actions, and legitimise the scenario imposed by the hostile group. That is, it would meet the opponents' expectations and confirm their prior accusations. The relationship with the Father, retained through the prayer, also helps to endure the non-illusory relationship with reality as a certain point of reference, while the reparation of the damage done by the supporters implements the inner consistency of that reality (Lk 22:51). This is why the lack of external support has a critical impact on the person's internal state, which is revealed by Jesus' attitude.

On the one hand, the victim's removal requires institutional support, which is achieved by influencing the decision-making person or persons, and on the other hand, there must be symbolic steps to confirm the legitimacy of the decision (Table 7).

The victim's physical removal becomes inevitable, but what matters is how it is received. A humble acceptance of reality and forgiveness protects against the disintegrating dynamics of psychological suffering, while a consistent attitude does not permit the full legitimisation of the institutional action. That is, the external mark of removal becomes a caricature, experiences a fiasco.

**Table 7.** Features of actions legitimising the victim's removal.

| Subcategory | Features | Reactions |
| --- | --- | --- |
| Impact on the decision-maker | Persuasion in hostility to the institution's authority (Lk 23:5, 5, 10), demonstration of the "majority's opinion" (Lk 23: 18, 21, 23). | A straight answer to accusations (Lk 23: 3). |
| Institution's behaviour | Opting out (Lk 23:7, 11), surrendering to the crowd (Lk 23:24), institutionally approved physical removal (Lk 23:24). | Acceptance of reality (Lk 23:3). |
| Justification | Marking shame and guilt (23:53), maintaining disbelief in righteousness (Lk 24:11–12; 21, 37). | Reasoning, explanation (Lk 24:24–26), evidence (Lk 24:39–43). |

## 5. Discussion

At first glance, the search for features of mobbing in the Gospel of Luke may seem controversial if one strictly adheres to the approach that the phenomenon is inherent solely to the workplace in the organisations of the industrial world. However, the review presented above and research on aggressive behaviour as a social phenomenon show that violence goes beyond the boundaries of the workplace (Hoel and Cooper 2001). Cunningham (1997), who has analysed the topic of persecution in the Gospel of Luke and Acts, emphasised that persecution here played a generally important theological function, highlighting the part of God's plan, related to the story of prophets, "was an integral consequence of following Jesus", and embodied Christian perseverance (Cunningham 1997). In this context, Christian spirituality is understood as a therapeutic resource, is based on discipleship or imitation, is aimed at strengthening the relationships between the individual, the community, and God (Frederick 2008, 2014), and therefore encompasses various areas of life, including work and interpersonal relationships wherein Christ's teaching and behaviour can be modelled. At the same time, it can be stated that Jesus' behaviour can be an appropriate model for dealing with workplace mobbing for a religious person.

Studies demonstrate that usually, after experiencing the first attacks, the victim of mobbing misjudges the situation, behaves passively without solving conflicts until they are removed (Shallcross et al. 2013, p. 191; Khoo 2010, pp. 62–63; Davenport et al. 1999, p. 38). This study highlighted the main types of responses to experienced persecution as a two-part model of behaviour: the first part is an external response to the attacks, and the second part involves internal processes (Table 8).

It should be noted that the criterion of publicity shows up in all actions. Similar to striving to publicly discredit the victim (Davenport et al. 1999, p. 33), the responses to verbal and non-verbal attacks described in the gospel are directed not only to the attacker but also to the audience that is watching. That is, "public opinion" remains important—attackers shape it seeking power and mobilising the support of the environment, while the right response does not make it possible to acquire that power. Therefore, publicity is considered an effective strategy (Davenport et al. 1999, p. 101). Lutgen-Sandvik (2006) explains how making negative actions public can serve the target of mobbing. Her study has demonstrated that bullying at work is always social and public; therefore, co-workers who are not stigmatised (thus, they retain "credibility" in the eyes of management) play a critical role in intervening in the dynamics of abuse. Meanwhile, hiding is an unsuccessful tactic. This way, the co-workers' support can reduce the power preponderance that a bully normally seeks to gain.

**Table 8.** The typical response to persecution actions in the Gospel of Luke.

| Kind | Type of Response | Actions of Persecution |
|---|---|---|
| Relationship with the outside | Reasoned explanation | Long-term persecution, false accusations, hostile interpretation, search for the enemy, justification of the institutional decision |
| | Disclosure of the truth | Institutional support and support of loved ones/ absence of support, impact on the decision-maker |
| | Insight and preventive actions | Long-term persecution, situation-trap, group conspiracy |
| | Disclosure of internal contradiction | Verbal violence, false accusations, crowd dynamics |
| | Public unmasking | Verbal violence, stigmatisation, conspiracy of the group, search for the enemy |
| | Dignified retreat | Non-verbal violence, fear of change |
| | Giving a sense to shameful "work" | Non-verbal violence |
| | Use of authority | False accusations, fear of change, situation-trap |
| | Transfer of moral responsibility to the opponent | Situation-trap, hostile interpretation |
| | Depersonalisation of the imposed image | Stigmatisation |
| Internal relationship | Forgiveness, love | Verbal and non-verbal violence, false accusations, group conspiracy, separation, experiencing of suffering |
| | Prayer, devotion to the will of the Father | Experiencing of suffering, temptation |
| | Meaning-giving, acceptance of reality while identifying oneself with the righteous | Stigmatisation, group conspiracy, separation, vertical and horizontal attack, institutional behaviour |

An active and non-confrontational behaviour in the Gospel of Luke does not allow for the imposed scheme to be fully implemented, and the identification with the group (the righteous who were persecuted) makes it possible to eliminate the negative effect of stigma (Whitson et al. 2017). Furthermore, a study by Noor et al. (2016) has shown that a person's higher level of spirituality plays a protective role in bullying when internalised stigmas threaten the victim's self-esteem. In general, external reactions here can be named as a consistent continuation of the internal coping with the experienced external impact. All the more so, as empirical research confirms that prayer, meditation (D'Cruz and Noronha 2010), and forgiveness help not only to overcome humiliation and suffering but also to eliminate negative emotional and cognitive consequences (Davenport et al. 1999, p. 65; Mishra et al. 2018; May et al. 2021). The study conducted by D'Cruz and Noronha (2012) has shown that the sense of injustice, experienced by victims of violence at work, particularly increases emotional suffering. Meanwhile, the results of another study have revealed a relationship between well-being and the feeling of healing with forgiveness reactions, regardless of their specificity (Mishra et al. 2018). In the narrative of the Gospel of Luke, forgiving the offender is based on the perception that he himself may not understand his actions, and the moral evaluation of the action is delegated to God (cf. Lk 23:34). This does not absolve one from responsibility for the wrong that has been done but changes the relation of the victim to the wrongdoer, as personal revenge is renounced. Following the example of the suffering Jesus, this can be named as a passive religious coping strategy, the efficiency of which is confirmed by other studies (e.g., Pargament et al. 2011); on the other hand, the example of forgiving Jesus for a religious person confirms the correctness of the chosen strategy. As stated by Bright et al. (2006), a transcendental moral framework provides a basis for forgiveness regardless of the context; therefore, forgiveness is viewed as a proactive practice that not only benefits the individual himself but also enables fully fledged interpersonal relationships.

The use of the biblical text in helping victims of workplace mobbing is enabled by practices used in Christian pastoral care. One of these can be the contemplative practice of Lectio Divina, which aims to allow practicing persons to engage in the narrative of the Holy

Scripture in their imagination and helps them to actively envisage God's active involvement in their experience (Frederick et al. 2021). For example, Greer et al. (2014) have found that Christian practices helped crime victims to forgive an insult. Although Lectio Divina was also tried along with other methods, the results in this respect were not singled out. However, a study by Jankowski and Sandage (2014) has demonstrated that contemplative prayer, which forms the basis for the Lectio Divina method, had a positive effect on the well-being of relationships through greater emotional and behavioural self-regulation. In this study, self-regulation was defined as the ability to observe one's emotional states, behaviour, and to adjust it according to desirable, prosocial ideals and goals. For a Christian, such an ideal is Christ, while discipleship is understood as overcoming difficult situations according to His example, which is directed to the final goal of humanity. Therefore, a sustainable internal attitude that does not lead to confrontation makes it possible to overcome the exerted pressure, to assess the situation objectively, and to efficiently respond to hostile behaviour, suppressing the escalation of the conflict. However, such model of response to workplace mobbing is only possible with a consistent religious belief that makes it possible to accept the hostile reality by knowing one's future (Lk 18:31–34) and perceiving the meaning of acting for a higher purpose (Lk 21:12–19).

## 6. Conclusions

To date, several studies have shown that when experiencing psychological violence at work, strategies such as prayer, meditation, and social support provision were useful in restoring emotional balance (D'Cruz and Noronha 2012), while spirituality acted as a protective mechanism that helped to preserve self-esteem (Noor et al. 2016). The aforementioned studies were conducted in non-Christian contexts (Hindu and Muslim); therefore, this article aims to outline guidelines that may help to integrate certain aspects of Christian spirituality into the processes of helping victims of mobbing. Workplace mobbing aims to eliminate the victim from a specific environment, using sophisticated measures that provoke the victim's feeling of helplessness and inability to resist, at the same time as forcing him/her to behave according to the planned scenario that legitimises the institutional decision to remove him/her. Jesus' persecution described in the Gospel of Luke corresponds to the typical features of mobbing; therefore, the evangelical narrative can be used in the process of Christian spiritual assistance for victims while structuring personal experiences. In this case, the Christian following of Christ as a model (e.g., Fil 2:5; Ef 5:1; 2 Kor 5:17; 1 Kor 11:1, etc.) can be applied both during the mobbing process itself and in dealing with its consequences after the removal from the organisation. On the one hand, it can be argued that the model of the victim's trauma and his/her approach to dealing with it is generally human, universal, and typical; thus, the search for analogies in the New Testament expands the opportunities of assistance by creating conditions for the approach of Christian spiritual assistance. On the other hand, the Gospel of Luke reveals the model of Jesus' response to the experienced attacks, which at the level of individual transactions helps to avoid the escalation of the conflict and the imposed self-destruction of the personality, and in the more general context, makes it possible to perceive and experience the meaning of personal experiences related to the violence of the people around. Answers to questions regarding how Christ behaved or would behave in one situation or another give the religious victim of mobbing an opportunity to look at the situation from a new perspective. What formally looks like an outrage, defeat, or a sign of weakness (e.g., 1 Kor 1:23), from the Christian perspective makes it possible to maintain a healthy relationship with reality while refusing the imposed illusion of helplessness. Of course, the most important limitation remains the qualitative dimension of the person's own faith, the purposeful strengthening of which requires additional pastoral efforts.

This study provides new knowledge and offers guidelines to clergy and laity working as religious counselors on how to apply the biblical text in the assistance process while helping to structure personal experiences, make sense of them in the context of faith, and make adequate decisions, avoiding further confrontation. This study also develops a

discussion of how religious resources can be applied to help religious victims of workplace mobbing. The categories of Christian response, distinguished in the study, provide new insights that can be used in future research, empirically investigating how the use of biblical texts can help the victims of mobbing to overcome difficult experiences and feelings.

**Author Contributions:** Conceptualisation, J.V. and M.D.; methodology, J.V. and M.D.; writing—original draft preparation, J.V. and M.D.; writing—review and editing, J.V.; supervision, J.V.; project administration, J.V.; funding acquisition, J.V. All authors have read and agreed to the published version of the manuscript.

**Funding:** This research received no external funding.

**Institutional Review Board Statement:** Not applicable.

**Informed Consent Statement:** Not applicable.

**Data Availability Statement:** Not applicable.

**Conflicts of Interest:** The authors declare no conflict of interest.

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
