# Peer review of "The New Testament and Workplace Mobbing: Structuring of Victims’ Experiences"

_religions, doi:10.3390/rel13111022_

Round 1

Reviewer 1 Report

Review Report

Article

 The New Testament and Workplace Mobbing: Structuring of 2 Victims’ Experiences

The purpose of this article is to correlate map of a workplace mobbing and mobbing in the Gospel of Luke. This study highlighted the main types of responses to experienced persecution. Starting point of the analysis is idea that spiritual help is an important resource in alleviating psychological and physical sufferings. The author analyzes selected biblical texts without a parallel in contemporary qualitative research, so recommendations for practice are debatable.

The structure of the introductory parts and the literature review parts identified the current state of the research question regarding of workplace mobbing and mobbing in the Gospel of Luke.

The cited references are from the older publications with some current ones (1988-2022).

Testability of the hypothesis: Analyzed research aims to form categories of typical patterns of mobbing behavior in sociological analysis and the Gospel of Luke. The analysis of the biblical text has the right method, but the research has no parallel in empirical research.

Features of mobbing in the Gospel of Luke (lines 169-267) are comprehensively processed.

Conclusions are too short (lines 309-333). Conclusion and findings are in the chapter 5 - Table 8: The typical response to persecution actions in the Gospel of Luke. It is very unfortunate that the results of this research are not compared with the results of potentially implemented empirical research results. The interpretation of the research results has taken place briefly in the light of the literature review, but not in the chapter 6 Conclusions.

The argumentation supporting the conclusion of an " Jesus’ persecution described in the Gospel of Luke corresponds to the typical features of mobbing; therefore, the evangelical narrative can be used in the process of Christian spiritual assistance for victims while structuring personal experiences.” (lines 313-314) is acceptable after improvement.

Conclusions are partially consistent with the evidence and arguments presented in the literature (for example: The answer to the Law Teacher’s question (Lk 10:25-28) as an excellent model of a non-confrontational reaction to a provocative situation.), but not with the research of workplace mobbing.

Quality and Scientific Soundness: The relevance of the review topic is supported by unused results of empirical research. The research question is original, but inconsistent.

I miss empirical research in this topic, especially in the area of the work place mobbing. Clarifying the questions in the qualitative empirical research. The manuscript’s results are only very generally reproducible.

I recommend the implementation of empirical research in the workplace today.

Significance: Hypotheses are roughly identified. The results of the sociological (not biblical) part are not collected; results from the biblical part are interpreted appropriately.  But their application into the current situation in the workplace mobbing is brief and indirect.

This paper may attract a selected readership.

Accept after Major Revisions: The paper can in principle be accepted after revision based on the reviewer’s comments.

Authors are given five days for minor revisions.

Author Response

Dear Reviewer,

We agree that the chosen biblical texts have not yet been analysed in the context of mobbing (this is also shown by the lack of direct empirical studies that would directly confirm the results of this study), and therefore we undertook this work in order to draw certain guidelines and promote discussions. We are very grateful for your valuable comments, which we tried to take into account and supplemented the manuscript.

We revised the research questions: 74-76.

We supplemented the discussion part with empirical studies: 347-353; 357-359; 364-379.

We supplemented the conclusions: 401-407; 432-440.

Sincerely

Authors

Reviewer 2 Report

As a sociologist, I found the purpose of this article quite confusing for a number of different reasons:

1. Since the article is based entirely on literature, and not on empirical research, there needs to be, in my view, a clear sense in which it is creating room for further study to take place. At present, the article seems to be pointing out some parallels between contemporary experiences of mobbing and stories about mobbing in the New Testament. For me, the big question that comes out of this is 'how do contemporary actors take up and make use of these stories? What does this look like in reality?' There seems to be a valuable project here, but this is not pointed towards strongly in the article.  

2. There needs to be a much better introduction to what 'mobbing' actually is, what it looks like in contemporary society, with some real life examples that illustrate instances of mobbing, even if these are taken from the existing research literature. The author presumes a familiarity with this area from the readers, and this does not help in the flow of the article and in its clarity.

3. The overall suggestion/argument, if I am right here, of the author is that the biblical stories (in Luke and elsewhere) are useful for victims of mobbing in processing and making sense of their experiences. But how they actually do this in real life was not clear. How do stories shape and change our own experience and give us guidance on how to act? This is surely a deep psychological question, and if the article is seeking to speak to psychology, then this is an area that needs to be considered in much more depth. Instead, the author seems to focus on the parallels between the types of mobbing present in the New Testament and experiences of mobbing in contemporary settings (giving us an almost categorising approach). But how are these stories taken up, and what help do they give to people in contemporary settings today? What does this actually look like? Perhaps because the paper was based entirely on literature, the author is not able to draw on original ethnographic or sociological research to be able to answer this question. If this kind of research has been carried out by the author, then why is it not included? 

4. I found the balance of tables and explanation in the article to be unhelpful. There seemed to be far too many tables categorising things and too little explanation of why these tables are significant in answering the question of how useful New Testament stories are to people today. My recommendation would be to cut back significantly on the tables, to summarise and explain them more in the text. 

5. The style of generalising in the article was also unhelpful, I think. For instance, as an example the author writes, "the main aim of mobbing is to remove the victim from the organization."  But are the aims of mobbing always the same? Surely there are lots of different types of mobbing? The lack of examples and illustrations of mobbing don't help in terms of us being able to assess this claim that the aim of mobbing is always to remove the victim from the organization. The author should avoid unsupported generalisations like this throughout the article. 

6. Finally, there are some phrases in English which need rewriting or clarifying. For instance, "Attacks on Jesus in the Gospel of Luke almost coincide with active activity"  What is meant here? And "the workplace is the only remaining “battlefield” where people can “kill” each other without fear of being convicted." Clarification is needed here that the word 'kill' is being used metaphorically. These are just a couple of examples. 

Author Response

Dear Reviewer,

Thank you for your work and comments, which we have taken into account and supplemented the manuscript:

  • We have explained how contemporary actors can take over and use biblical narratives: 326-330; 380-394.
  • We have additionally provided an example from an auto-ethnographic study, illustrating mobbing in the workplace: 79-94.
  • We have added an explanation of what is meant by “active activity”: 215-217.
  • We have added authors who name the removal of the victim as a goal: (Pheco, Leymann...): 165-166.
  • We have also supplemented the conclusions: 432-440.
  • The word “kill” is used by our quoted Leymann, and the quotation marks indicate reference to a figurative, metaphorical meaning; therefore, additional explanation would be redundant. Thanks for the opinion, but we think it makes sense to leave the tables that may be useful to practitioners, as they provide references to specific lines of text, which can be used in providing assistance.

Sincerely

Jolita Vveinhardt

Reviewer 3 Report

I really enjoyed reading your paper. The topic about which you chose to focus is fascinating and important. Thanks for a good manuscript. I do have a few suggestions for you with regard to enhancing your manuscript:

1) Please provide a brief definition of mobbing earlier in the manuscript. I was not clear about what it was that you were discussing until the section starting on line 64.

2) I was unclear what you meant in lines 46 and 47 when you mentioned the context of Christian pastoral care. It sounds like you are suggesting that mobbing happens in that context. Or, were you suggesting that the results of your study could be beneficial in Christian pastoral care?

3) Were the "attacks on Jesus" (line 169) your primary focus in this project? If so, that should be made clearer prior to when you mentioned it in line 169.

4) My primary concern is with understanding your method. Is this an analysis of existing research/scholarly literature that led to the development of some categories that you used when analyzing the biblical narratives. Was your analysis of the biblical narratives a content analysis? If so, please describe a bit more about how that analysis was conducted.

Author Response

Dear Reviewer,

Sincerely thank you for your positive review. Considering your proposals, we have made several improvements:

  • we have inserted the definition of mobbing in the introductory part: 36-40;
  • we have removed the ambiguity: 58-60.
  • we have expanded the description of the research methodology: 179-178; 186-199.

Sincerely

Jolita Vveinhardt

Reviewer 4 Report

There is no doubt,as the Author of the article writes, that: "At first glance, the search for features of mobbing in the Gospel of Luke may seem controversial if strictly adheres to the approach that the phenomenon is inherent solely to the workplace in the organizations of the industrial world. However, the review presented above and research on aggressive behaviour as a social phenomenon show that violence goes beyond the boundaries of workplace". The problem, however, is that the theological analysis of the Gospel is hardly available to Christians who declare their deep religious faith. The application of the content of the Gospel to the interpretation of everyday events belongs to theologians rather than to Catholics. It can be agreed that the text is intended for priests who provide psychological advice and help the faithful to solve their mental problems. On the other hand, psychological advice is rarely provided by priests (sociological research in Poland). Therefore, I propose to supplement the text with indications to which reader the article is addressed and how the conclusions from the analysis linking the content of the Gospel with mobbing can be used in pastoral work.

The text contains inconsistent records of Gospel passages (LK 6,3-5; LK 6:-10) and the omissions on page 5 (LK 26:.............) It is necessary to unify and supplement the provisions.

Author Response

Dear Reviewer,

Thank you very much for your observations and comments. The article is intended for counsellors who are both clergy and laity. We have therefore made additions and eliminated shortcomings: Table 2 and elsewhere; 380-394, 432-440 lines.

Sincerely

Jolita Vveinhardt

Round 2

Reviewer 1 Report

Conclusions are in the corrected version consistent with the evidence and arguments presented in the literature. 

Application into the current situation in the workplace mobbing is specified and usable for further research.

Reviewer 2 Report

Thank you for the reviews that you have made to the manuscript. The purpose and clarity of the paper have been improved.